

# Supraglacial pond evolution in the Everest region, central Himalaya, 2015-2018.

Caroline J. Taylor[1], J. Rachel Carr[1]

[1]School of Geography, Politics and Sociology, Newcastle University, Newcastle Upon Tyne, NE1 7RU, UK.

*Correspondence to*: Caroline J. Taylor (C.Taylor11@newcastle.ac.uk)

**Abstract.** Supraglacial ponds are characteristic of debris-covered glaciers and greatly enhance local melt rates. They can grow rapidly and coalesce to form proglacial lakes, which represent a major hazard. Here, we use Sentinel-2A satellite imagery (10 m) to quantify the spatiotemporal changes of 6,425 supraglacial ponds for 10 glaciers in the Everest region, Nepal, between 2015 and 2018. During the study period, ponded area increased on all glaciers, but showed substantial temporal and spatial variation, and the rate of pond growth increased substantially relative to 2000-2015 (Watson et al., 2016). Both Imja and Spillway Lake expanded and Khumbu Glacier developed a chain of connected ponds. 54% of ponds were associated with an ice-cliff, but the proportion of ponds with cliffs decreased during the study period. Pond location generally corresponded to lower surface velocity, but this relationship was not ubiquitous. Ponds are now predominantly found at mid-elevations on our study glaciers, suggesting that conditions conducive to pond formation have advanced up-glacier compared to general theory. Results demonstrate the need to utilize high-resolution imagery (< 10 m), as using Landsat (30 m) would miss 55–86 % of the total ponds. Glaciers were classified by stage of development (Komori, 2008; Robertson, 2012). Two glaciers transitioned between stages between 2015 and 2018, suggesting that lakes in the region are evolving rapidly. Some study glaciers displayed characteristics of multiple classes, so we propose an adapted classification system. Overall, our results demonstrate rapid pond expansion in the Everest region and highlight the need for continued monitoring for hazard assessment.

*Key Words: Supraglacial ponds, remote sensing, outburst floods, hazards.*

## 1 Introduction

Mass loss from glaciers in the Himalayas has increased rapidly over the past 30 years, in response to climate change (e.g. Bolch et al., 2012; Kääb et al., 2012; Quincey et al., 2009; Gardelle et al., 2013). Here, 'summer-accumulation type' glaciers rely on summer-monsoon snowfall for mass gain (Bolch et al., 2012), which is thought to be reducing as temperatures rise (Fujita, 2008). The shrinkage of these freshwater reservoirs will have significant regional- and local-scale impacts (Immerzeel et al., 2010; Bolch et al., 2012; Dehecq et al., 2018). Seasonal monsoonal rainfall is the dominant source of water in the Himalaya, but glacial meltwater provides up to 40% of water supplies during the dry season (Immerzeel et al., 2010). Thus, the consistent negative mass balance of Himalayan glaciers may lead to long-term reductions in perennial flow supplied to



major rivers outside of monsoon season (Xu et al., 2009), and could threaten the water and food security of an estimated 70 million people in the densely populated downstream catchments (Immerzeel et al., 2010). Additionally, increased supraglacial meltwater storage will likely increase the frequency of glacier related hazards in the region, particularly from Glacier Lake Outburst Floods (GLOFs) (Thompson et al., 2010). GLOFs are highly destructive and Nepal (along with Bhutan), has been identified as the most economically vulnerable to these hazards (Carrivick and Tweed, 2016), making it vital to assess how the

GLOF risk will evolve with climate warming.

   The Everest region comprises three catchments spanning the Nepal/Tibet border in Eastern Nepal (Fig. 1; King et al., 2017). These catchments have experienced atmospheric warming since the mid-1970's (Shrestha et al., 1999; Shrestha and Aryal, 2011) and weakened summer monsoons, which has reduced glacier accumulation (Salerno et al., 2015). Consequently, glacial mass balance in the region has been strongly negative since the 1970s (Bolch et al., 2008; Benn et al., 2012; Kääb et al., 2012).

For example mass balance in the Everest Region of Nepal was -0.22 ± 0.12 m w.e.a$^{-1}$ between 1999 and 2011 (Gardelle et al., 2013). Glaciers in the Everest region are characterised by ice-surface rock debris, which is sourced from the surrounding hillslopes and covers approximately ~80% of the glaciated area in the region (Fushimi et al., 1980; Sakai et al., 2000; Watson et al., 2016). The presence of this surface debris substantially alters the glacier mass balance gradient, in comparison to clean-ice glaciers: melt is supressed close to the termini, due to the presence of thick debris which insulates the underlying ice, and

enhanced up glacier, where debris is thinner and increases melt (e.g. Quincey et al., 2009; Nicholson and Benn, 2013). As a result, the glaciers are characterised by high elevation accumulation areas, and lower-elevation ice tongues, which have low surface slopes and are near-stagnant, due to the low driving stresses (e.g. Bolch and Kamp, 2006; Quincey et al., 2009; Bolch et al., 2011; Dehecq et al., 2015). Consequently, glaciers in the Everest region lose mass by widespread surface lowering (i.e. down-wasting), rather than through terminus retreat (Quincey et al., 2007; King et al., 2017).

The low surface slopes and slow ice velocities that characterise debris-covered glacier tongues in the Everest region (Quincey et al., 2007; King et al., 2017), and the Himalaya more broadly (e.g. Kääb, 2005) facilitate supraglacial pond formation, by allowing glacial meltwater and rainfall to accumulate in depressions on the glacier surface (Reynolds, 2000; Sakai et al., 2000; Miles et al., 2016 and 2017). These ice surface lakes can then coalesce to form a larger proglacial lake, which carries the risk of producing GLOFs (e.g. Richardson and Reynolds, 2000; Quincey et al., 2007; Thompson et al., 2012; Mertes et al., 2017).

Supraglacial ponds are highly variable in character (e.g. in shape, size, turbidity and ice-cliffs), dynamic in nature, and are expected to become increasingly prevalent in a warming climate (Thompson et al., 2016; Miles et al., 2017; Watson et al., 2016; 2017a and 2017b).Understanding the spatial and temporal patterns of pond growth is therefore vital for accurately forecasting proglacial lake growth and the associated hazard of GLOFs (Richardson and Reynolds, 2000; Quincey et al., 2007; Benn et al., 2012).

As well as posing a hazard through the formation of proglacial lakes, supraglacial ponds represent hotspots for ablation on low-gradient debris-covered tongues (Miles et al., 2017; Watson et al., 2017a and 2017b). This is because they have a comparatively low albedo and therefore absorb more insolation, which they transmit to surrounding ice, resulting in higher melt rates (e.g. Reynolds, 2000; Benn et al., 2001; Röhl, 2008; Miles et al., 2016; Watson et al., 2016; Mertes et al., 2017;





Salerno et al., 2017). These enhanced melt rates cause the ponds to expand both horizontally, through subaerial and sub-
aqueous melting at the margins, and vertically, via basal melting (e.g. Sakai et al., 2000; Röhl, 2008; Mertes et al., 2017).
Variations in the spatial distribution of ponds across a glacier is thought to be governed by surface slope and velocity (e.g.
Reynolds, 2000; Bolch et al., 2008; Quincey et al., 2009), and results in differential surface melt rates (Sakai et al., 2000; Benn
et al., 2001; Miles et al., 2016; Watson et al., 2016; Mertes et al., 2017). Once formed, ponds can persist for months to years,
or drain via englacial pathways (e.g. Immerzeel et al., 2014; Miles et al., 2017; Watson et al., 2017a). Patterns of pond drainage
are generally governed by their interaction with crevasses and englacial features, which provide efficient drainage outlets
(Benn and Lehmkuhl, 2000; Miles et al., 2017). Determining whether ponds drain regularly or persist is important, because
long-duration ponding can cause substantial ice-surface melt, whereas repeated pond drainage events can convey energy into
the glacier's interior (Miles et al., 2017), resulting in quite different ice loss patterns. Additionally, recent studies (e.g. Brun et
al., 2017; Buri et al., 2016; Mertes et al., 2017; Watson et al., 2017b) suggest that ice-cliffs play a significant role in pond
formation, by enhancing marginal pond melt and subaerial calving. Furthermore, as ponds expand, ice and debris influx into
the pond from retreating cliff-tops increases, causing pond turbidity to increase and subsequently reducing albedo, initiating a
positive feedback of melt (Mertes et al., 2017). These factors complicate predicting future supraglacial pond formation and
evolution and make them highly dynamic features.

Previous studies have documented changes in supraglacial water storage across the Himalaya (Table 1; e.g. Wessels
et al., 2002; Kattlemann, 2003; Bajracharya and Mool, 2007; Gardelle et al., 2011). However, the use of comparatively coarse
resolution imagery (e.g. 30m resolution Landsat imagery and 15m resolution ASTER imagery) means substantial water
volumes may be missed and the low repeat frequency makes differentiating between pond persistence and regular drainage
difficult, which is important for quantifying the impact of the ponds on mass loss (Immerzeel et al., 2014; Miles et al., 2017).
Watson et al. (2016), presented the first high temporal and spatial resolution study of supraglacial pond evolution in the Everest
region, for the period 2000-2015, where a total of 9340 ponds were identified. Here, we extend this previous work to quantify
supraglacial pond evolution between 2015 and 2018. This will provide an up-to-date picture of pond coverage within the
region. Our main objectives are; (1) characterise the spatial and temporal evolution of supraglacial ponds, to determine the
magnitude and extent of change since 2015, (2) assess the impacts of using higher resolution imagery versus lower resolution
imagery on the ability to identify and assess supraglacial ponds (3) examine the impact of local glacier characteristics (glacier
surface slope, ice velocities and the presence of ice cliffs) on pond formation location, area and number and (4) classify the
stage of proglacial lake development across the region.

## 2 Methods

### 2.1 Study Site and Water-Body Definitions

The study focuses on ten debris-covered glaciers that drain the Dudh Koshi basin, within Sagarmatha National Park, Eastern
Nepal (Fig. 1). The glaciers flow predominantly in a southerly direction, with the exception of Ama Dablam (north flowing)





and Imja (west flowing) glaciers (Fig. 1). All of the study glaciers have extensive debris-covered tongues and high accumulation areas. Ngozumpa, Pangbung and Khumbu glaciers have the greatest length (~15 km, ~13 km and ~11 km respectively) and Imja Glacier the shortest at ~2 km (Table 2). Of the 10 glaciers in this study, 8 were also included in Watson et al. (2016) study for the period 2000-2015 (Table 2). We add Pangbung Glacier and Sumna Glacier to our study, as they are

important for downstream water resources and could potentially pose a threat to downstream communities due to GLOFs (Immerzeel et al., 2010). The distinction between glacial 'pond' and 'lake' remains poorly defined in the Himalayan literature, with Watson et al. (2016) referring to all surface water as ponds unless specifically named otherwise, and other authors switching between the terms 'lake' and 'pond' (e.g. Gardelle et al., 2011; Nie et al., 2013). Here, we use the term pond to refer to all bodies of water on the glacier surface. Proglacial lakes are discussed separately from ponds, and are defined as all water

bodies that are outside of the glacier margin, but are in contact with it. Following this classification, 'Spillway Lake', located at the terminus of Ngozumpa Glacier, is classified as a pond as it remains bound by glacier ice on all sides, but here is discussed separately, as its very large area would skew results. Imja Lake, located at the front of the Imja/Lhotse Shar glacier complex is classed as a proglacial lake.

## 2.2 Data Sources

This study used true-colour orthorectified Sentinel 2A imagery (<10 m resolution) (available from USGS at http://earthexplorer.usgs.gov/) for the time period December 2015 to April 2018 (SI. Table 1). Images were chosen outside of the monsoon season (June-September), to minimize cloud cover (see SI. Table 1 for percentage cloud cover), and, where possible, images were selected from the same month to avoid inadvertently including seasonal differences into our analysis (SI. Table 1). For 2016-2018, we used imagery from April, but for 2015, we had to use data from December (SI. Table 1), as

Sentinel 2A data are only available from November 2015 and we wanted to ensure that our study period continued directly on from that of Watson et al. (2016), to facilitate comparison. Furthermore, our image dates (April and December) are outside of the monsoon and summer season, which should minimise seasonal effects. True colour images were derived from combining the blue (490 nm), green (560 nm) and red (665 nm) bands. For each of the study years, two true-colour Sentinel-2A images from the same date were mosaicked to produce one spatially continuous dataset of the entire region. Glacier outlines were

obtained from the Randolph Glacier Inventory 5.0 (available from GLIMS at http://www.glims.org/maps/glims) and modified manually to reflect the debris-covered area of each glacier. These were then used as glacier / land masks and only ponds located within this mask were included in the study.

## 2.3 Maximum Likelihood Classification and Manual Editing

A supervised classification technique was used to automatically delineate supraglacial ponds. First, we manually selected

training sites that contained the primary land cover classes (e.g. clean ice, water, debris covered ice etc.) from the true-colour Sentinel 2A image. We then performed the Maximum Likelihood Classification (MLC) on the true-colour image, plus bands 5 and 7 (near infrared and thermal wavelengths respectively): both infrared and thermal wavelengths are absorbed by water



bodies so their addition aided the classification substantially. Whilst other classifications can be used (e.g. the Hierarchical Knowledge Based Classifier (HKBC) method), previous studies suggest that MLC is the most accurate classification method

for delineating water stores on glaciers (Tiwari et al., 2016). This method was repeated for all image dates. The classification results were assessed manually, by comparing automatically detected pond margins to the underlying imagery. We then manually edited any ponds where the classification had failed to accurately detect the pond margins. In total, we identified 6,533 ponds, and then extracted key statistics for analysis, specifically ponded area and number of ponds.

### 2.4 Controls on Pond Location

We assessed controls on supraglacial pond formation and growth patterns, specifically: glacier surface slope, ice velocities and the presence of ice cliffs. We derived a slope map and glacier elevation profiles from the ASTER DEM ( http://earthexplorer.usgs.gov/) to identify glaciers with particularly low slope gradients and thus potential areas for future pond development. Glacier velocities were derived from repeat-image feature tracking of Landsat images by Dehecq et al. (2015). The data used in this study are derived from average velocities for 2013-2015 and have a 120 m spatial resolution (Dehecq et

al., 2015). Each of the study glaciers were divided into 10 bands, representing 10% glacier surface area, to facilitate comparison of pond locations with surface slope and ice velocities. Ice cliffs were mapped manually using the true-colour Sentinel-2A images for all time periods.

## 3 Results

### 3.1 Supraglacial Pond Change

#### 3.1.1 Regional Ponded Area Change

Across the study region, total ponded area increased on all 10 glaciers between 2015 and 2018 (Fig. 2). The most prominent changes in both pond number and area were observed on the three largest glaciers (SI. Table 2; Ngozumpa, Pangbung and Khumbu glaciers), which contain ~58% of the total 6,533 ponds identified in this study (Fig. 2). Specifically, ponded area increased by 255,849 m$^2$ (33.3 %) on Ngozumpa Glacier, 191,386 m$^2$ (36.2 %) on Pangbung Glacier and 134,299 m$^2$ (43.1 %)

on Khumbu Glacier between December 2015 and April 2018 (Fig. 2). This increase mainly resulted from pond coalescence and growth on Ngozumpa and Khumbu glaciers, as the number of ponds decreased but the ponded area increased (Sl. Fig. 1): the number of ponds decreased by 60 (13.5% decrease) on Ngozumpa and 3 (1%) on Khumbu (Fig. 2.). The remaining seven smaller glaciers showed an increase in ponded area between 2015 and 2018, ranging from 9,664 m$^2$ on Imja Glacier (30 % increase) to 120,162 m$^2$ on Lhotse Glacier (68 % increase). This increase in area was a result of both the establishment of new

ponds and the coalescence of existing smaller area ponds (Fig. 2), for instance ponded area on Ama Dablam Glacier increased 38,958 m$^2$ and pond number increased by 29, demonstrating new pond formation. Generally, the ponded area increased on all 10 study glaciers, but most marked on the three larger glaciers.





Whilst total ponded area on all glaciers increased from 2015-2018 (SI. Table 2), the number of ponds found on each glacier varied (Fig. 2). Only 2 of the 10 glaciers showed an increase in pond number: on Pangbung Glacier the number of

ponds increased by 4(200 ponds total), and on Ama Dablam Glacier the number of ponds increased by 29(67 ponds total) between December 2015 and April 2018. The remaining 8 glaciers exhibited an overall decrease in the number of ponds, ranging from a reduction of 2 (Lhotse Glacier) to 60 (Ngozumpa Glacier). Glaciers with decreasing pond numbers generally showed an increase in ponded area in 2015, 2017 and 2018 (SI. Fig. 1). The exception to this inverse relationship between pond number and ponded area was in 2016 (SI. Fig. 1). During this year, a large decease in pond number was observed on

glaciers in the eastern part of the study area, particularly Lhotse, Lhotse Shar and Lhotse Nup glaicers (Fig. 2b). At the same time, there was a large increase in pond number on glaciers in the western part of the study area (Fig. 2c). For example Pangbung Glacier increased in pond number from 196 (2015) to 468 (2016), and Sumna Glacier increased from 78 (2015) to 182 (2016). Overall, our results show that ponded area increased on all glaciers during the study period 2015-2018, whereas the number of ponds generally declined, despite the considerable spatial and temporal variability (Fig. 2).

Two water bodies in the study area were assessed separately, due to their large size; supraglacial lake 'Spillway Lake' on the terminus of Ngozumpa Glacier, and proglacial lake 'Imja Lake' fronting the Imja/Lhotse Shar Glacier complex (Fig. 4). Spillway Lake, located at the terminus of Ngozumpa Glacier, underwent a net gain of 40,565 m² during the study period (December 2015 - April 2018) and reached a size of 286,367 m². This remains the largest surface water store in the study region. The only proglacial lake identified in this study was Imja Lake, which expanded up-glacier by 455.6 m to reach and

area of size of 1,493,142.68 m$^2$ by 2018. Whilst these two water bodies are currently the only very large water bodies in the study area, our data show substantial growth and coalescence of surface ponds on Pangbung and Khumbu glaciers (Fig. 3b, c).

### 3.1.2 Glacier-scale Pond Changes

Despite the overarching trends across the region, changes in supraglacial pond number and area varied from glacier to glacier

and across individual glaciers (Fig. 2; Table 2). For the largest three glaciers, most of the ponded area was located near the terminus, and this persisted throughout the study period (Fig. 3a-c). The number of ponds, although demonstrated an overall decrease, showed limited year to year variation, whilst ponded area increased (Fig. 2), which was primarily as a result of pond coalescence on the lower glacier tongues. (Fig. 3a-c). For example, on the terminus of Ngozumpa Glacier, two smaller ponds (Fig. 3a, i and ii) formed new branches of Spillway Lake in April 2017. This reduced the total number of ponds by 2, whilst

increasing the total ponded area of Spillway Lake. Similarly, along the eastern margins of Khumbu Glacier, lateral pond expansion between April 2016 and April 2017 (Fig. 3b, i and ii) resulted in coalescence of two major ponds and a reduction in the number of surface ponds by 2. Our data therefore suggests that supraglacial pond expansion on the larger glaciers in our study area predominantly results from pre-existing ponds coalescing, rather than by the formation and growth of new ponds.

On the seven smaller glaciers, there was substantial spatial and temporal variability in the number and area of surface

ponds, both between the glaciers and across individual glaciers (Fig. 3d and Fig. 4). For example, Lhotse Shar and Imja Glacier



neighbour each other, in the east of the study region (Fig. 1). However in April 2018, Lhotse Shar Glacier had over three times the ponded area (135,420 m$^2$) of Imja Glacier (42,603 m$^2$) as well as almost three and a half times the number of ponds (96 and 28 respectively). Sumna Glacier in the west of the region showed major variations in ponded area and number over time, as its percentage pond cover ranged from 2.16 % to 4.87 % during the four-year study period. For most of the smaller glaciers,

ponds occupied similar locations at each time step (e.g. Ama Dablam Glacier SI. Fig. 2a and Lhotse Glacier SI. Fig. 2b). Sumna Glacier was the exception to this and showed a distinctive change in the spatial pattern of the pond locations during the study period: at the start of the study (2015), it had more ponds at higher elevations, but by the end (2018), ponds were most concentrated near the terminus (Fig. 3d). Overall, our results show an increase in both ponded area and number of ponds on the smaller glaciers, but this showed substantial spatial and temporal variation, even between neighbouring glaciers and on the

same glacier.

### 3.2 Controls on Pond Location

### 3.2.1 Glacier Elevation Profile

The study glaciers have an average slope >18° at the tongue and <25° at higher elevations. In general, areas where the mean

overall slope is lower (>10°) contain more ponds Overall, slope angle tends to decrease closer to the glacier termini, where slopes are generally between 2° and 4° (Fig. 5). However, with the exception of Sumna Glacier (see section 3.1.2.) there is no apparent relationship between elevation and pond number/ area on the study glaciers (Figs. 5 and 6). We assessed this in further detail by dividing each glacier into 10 equal elevation bands (to account for differences in total length and to facilitate direct comparison between glaciers) and calculating the number and area of ponds in each elevation band (Figs. 5 and 6): Bands are

numbered from 1 (the glacier head wall) to 10 (terminus). On six of the ten study glaciers, the number of ponds decreased from the head of the glacier to terminus, whilst the pattern was reversed on the remaining 4, so that pond numbers increased with distance up glacier (Fig. 5 and 6). Furthermore, all the study glaciers showed large variations in ponded area and number between individual elevation bands.

For all study glaciers, the largest number of ponds were usually found in the central elevation bands (bands 5-6), as

exemplified by Pangbung (32.2 % of ponds), Khumbu (23.1 %) and Lhotse Shar (47.8 %) glaciers (Fig. 5). The exception to this trend can be seen on Imja, Lhotse and Sumna glaciers, where the highest numbers of ponds can be found nearer to the termini (i.e. elevation bands 9 and 10), and Ama Dablam Glacier where most ponds are located nearer the high accumulation zone (bands 1-2; Fig. 5). Furthermore, the number of ponds was much higher on Imja Glacier's terminus (band 10; 6 ponds; 21.4 %) than on any of the other study glaciers. No ponds where identified in the high accumulation zone (band 1) for four of

the ten study glaciers; Imja, Nuptse, Lhotse Shar and Sumna (Fig. 5 and 6). Sumna Glacier also displayed distinctive areas of higher frequency ponding (bands 6-8) and lower frequency ponding (bands 2-4) which showed no clear relationship with elevation (Fig. 3d; Fig. 5). A number of breaks in slope were identified on six of the ten study glaciers: Ama Dablam, Imja,



Lhotse, Lhotse Shar, Lhotse Nup and Sumna (Fig. 5). In the band immediately down glacier of the break in slope, there was
an increase in pond number on four of the glaciers (Imja, Lhotse, Lhotse Shar, and Sumna; Fig. 5). This was most notable on
Lhotse Glacier, where there was a change in slope in band 1 and the number of ponds increased from 1 (band 1) to 20 (Band
2) (Fig. 5). Ama Dablam and Lhotse Nup were the exception to this trend, where on Ama Dablam Glacier there was a decrease
in pond number from 8 to 4, following the break in slope in band 3 (Fig. 5) whilst on Lhotse Nup a break in slope in band 6
preceded a decrease in pond number from 7 to 1 (Fig. 5).

### 3.2.2. Glacier Velocity

Generally, glacier velocity decreased from source to terminus, being highest in bands 1-3 and lowest in the bands 8-10 (Fig.
7). Where velocities were higher, the number of surface ponds was generally lower (Fig. 7). For instance, in bands 1 and 2 on
Lhotse Shar Glacier there were no ponds recorded (velocity >20 ma$^{-1}$). However, further down glacier from band 3, velocities
were lower (<20 m a$^{-1}$) and the number of ponds increased by 14 (14.6 %) (Fig. 7). The main exception to this trend is Nuptse
Glacier, where velocity is low in bands 1-2 (> 2 ma$^{-1}$) and there are no ponds, but the area of higher velocity in bands 4 to 5
(>14 ma$^{-1}$) contains a total of 34 ponds (Fig. 7). In general, where velocities are lowest, for instance nearer the glacier terminus,
pond number increases, (Fig. 7). For example, from band 5 onwards on Lhotse Shar Glacier, there is almost no recordable
velocity and pond number reaches its highest (23 ponds) (Fig. 7). The relationship between total pond area and glacier velocity
is similar to that for pond number (Fig. 8): higher velocities coincide with lower ponded area, whereas lower velocities have a
higher ponded area (Fig. 8). Sumna Glacier displays this relationship clearly: it has 23 % of its ponds in Bands 2-4 and 70 %
in bands 6-8. This corresponds to velocities of 3 to 7 ma$^{-1}$ and < 3 ma$^{-1}$ respectively. Overall, our results indicate that lower
velocities correspond with higher ponded area and pond number, and higher velocities generally relate to fewer ponds.

### 3.2.3. Ice Cliffs

Although ice cliffs were identified on all 10 glaciers 2015-2018, the number of cliffs showed marked spatial and temporal
variation during this period (Fig. 9). Ngozumpa and Khumbu glaciers had the highest percentage area of the glacier covered
by ice cliffs, with 4.3% and 3.92% respectively, and Sumna glacier the least (1.1%; Fig. 9). The greatest temporal variability
was observed on Ama Dablam Glacier, where percentage ice-cliff coverage decreased from 2.59 % in 2015 to 1.3 % in 2016,
and then rapidly increased to 3.3 % in 2018 (Fig. 9). The number of supraglacial ponds with a corresponding ice-cliff exceeded
the number without: on average, across all of the study glaciers, 54 % of ponds had a coincident ice-cliff (Fig. 9). During the
study, the number of ponds without an ice cliff increased on average by 1.6 % of the total glacier surface area and this was
most noticeable on the seven smaller glaciers (Fig. 9). For instance, on Sumna Glacier, the area of the ponds with a cliff
remained relatively stable (~ 0.8 %), whereas the area of ponds without a cliff increased from 1 % in 2015 to 1.75 % in 2018
(Fig. 9). In comparison, the number of ponds with an ice cliff increased by just 0.9 % (Fig. 9). For example, on Ngozumpa
Glacier, ponds with cliffs increased by 0.84 % over the four year period compared to 1.54 % over the four year period for



ponds without cliffs. This may indicate pond growth can occur irrespective of ice cliff presence, thus whilst ice cliffs continue

to form in the Everest region, ponded area is increasing at a greater rate.

### 3.3 Future Lake Development

Each of the 10 study glaciers was assigned a number, according to the stage of lake development described in established lake classification schemes (Komori, 2008; Robertson, 2012; Table 3). In 2015 (Fig. 10b), the study region was dominated by glaciers in Stage 2 of lake development, with 60% showing ponds that have coalesced, were ice-dammed and have large

ponded areas (>20,000m$^2$). Only Ngozumpa Glacier was defined as Stage 3, due to the presence of the large terminal Spillway Lake (Fig. 10). The remaining three glaciers (Sumna, Lhotse Nup and Ama Dablam Glaciers) were all classified as Stage 1, with supraglacial ponds forming in their lower ablation zones (Fig. 10b).

Over the four-year study period, (December 2015- April 2018) two glaciers (Ama Dablam and Lhotse Nup) transitioned to a new stage of lake development (Fig. 10). Both progressed from Stage 1, where a few supraglacial ponds were identified, to

Stage 2, where ponds had begun to coalesce (Fig. 10). Two glaciers (Pangbung and Lhotse Shar) partially transitioned from Stage 1 to Stage 2, and from Stage 2 to Stage 3, respectively (Fig. 10). Features that fit more than one stage were identified on these two glaciers, meaning that they could not be assigned a stage using the current classification. For example, on Pangbung Glacier, ponds were appearing on the lower ablation zone (characteristic of Stage 1), but some ponds were also beginning to coalesce (Stage 2; Fig. 3c). On Lhotse Shar Glacier, coalescing was observed (Stage 2), but there was also stable expansion of

its proglacial lake (Stage 3). As a result, the current proglacial lake development in the study area cannot be captured by existing classification schemes.

### 4 Discussion

### 4.1. Changes in Supraglacial Pond Area 2015-2018

The area of supraglacial ponds in the study increased markedly between 2015 and 2018, ranging from a 13.6 % increase on

Sumna Glacier to a 108.1 % increase on Lhotse Nup Glacier, despite showing large inter-annual variations (Fig. 2). These increases show a marked acceleration in pond growth, compared to 2000-2015 (Watson et al., 2016). For instance, ponded area increased at a rate of 1.6 % a$^{-1}$ on Nuptse Glacier between 2000 and 2015 (Watson et al., 2016), whereas our study measured a rate of 22.6 % a$^{-1}$ between 2015 and 2018 (SI. Table 2). Similarly, the rate of expansion on Lhotse Glacier in our study (17.0 % a$^{-1}$) is four times greater than that found by Watson et al. (2016; 4.1 % a$^{-1}$). This is a major concern in terms of

risks to downstream communities, as these very high rates of pond growth will rapidly increase the water volumes available for outburst floods and will also encourage pond coalescence and the rate of proglacial lake formation.

One potential explanation for the observed acceleration in pond growth relates to climatic controls: warmer air temperatures should increase melt rates and hence encourage pond expansion, whilst increased precipitation could add water directly to the ponds. Data on climate trends proximal to our study glaciers are very limited. However, available data (Salerno





et al., 2015) suggest that minimum and mean air temperatures have risen in the Everest area between 1994 and 2013, at elevations above 5000 m. However, warming was most marked in spring and winter, and would thus have a more limited impact on ice melt, and it was concurrent with a reduction precipitation (Salerno et al., 2015), which would decrease direct inputs to the ponds. As such, we suggest that the observed increase in ponding may at least partly reflect changes in the dynamics of our study glaciers, which may provide the conditions that promote pond formation. Between 2000 and 2017,

glaciers in East Nepal decelerated by -1.8 ± 0.1 ma$^{-1}$ (17.0 ± 1% a$^{-1}$) and thinned, which in turn reduced driving stresses (Dehecq et al., 2018). Down-wasting and deceleration creates an inverted mass balance gradient and an uneven glacier surface, which together facilitate pond formation (e.g. Reynolds, 2000; Miles et al., 2016 and 2017). Furthermore, slow flow is likely to reduce the number of crevasses forming and would thus reduce the chance of pond drainage (Immerzeel et al., 2014; Miles et al., 2017; Watson et al., 2017a and 2017b). As such, we suggest that recent changes in ice dynamics in the Everest region

are likely to be contributing to observed pond growth and that this may lead to a positive feedback, whereby rapid pond growth accelerates down wasting, leading to further pond expansion.

## 4.2. Glacier-scale Ponded Area Patterns

The three larger glaciers (Ngozumpa, Khumbu and Pangbung) increased substantially in ponded area between 2015 and 2018 (Fig. 2: 33.3 % on Ngozumpa Glacier, 36.2 % on Pangbung Glacier and 43.1 % on Khumbu Glacier). Changes in pond number

show less net change, with only Ngozumpa Glacier demonstrating a clear decline in pond number (decrease of 60) (Fig. 2). This suggests that pond area is generally increasing via the expansion of existing ponds and through pond coalescence, as seen at the terminus of Ngozumpa Glacier (Fig. 3a) and the eastern margins of Khumbu Glacier (Fig. 3b), with new pond formation playing a smaller role, accounting for the year to year variability. This has implications for both glacier lake related hazards and ice loss rates in the future. As ponds continue to join, and the area of supraglacial ponds increases, the likelihood of

proglacial lakes formation is increased (Komori, 2008; Robertson, 2012), which in turn increases the risk of GLOFs (e.g. Richardson, 2000; Quincey et al., 2007; Benn et al., 2012; Rounce et al., 2017). In addition, the larger ponded surface area increases the wind fetch, which may lead to enhanced undercutting at pond margins and thus increase glacier melt rate (Benn et al., 2001; Röhl, 2006, 2008; Sakai et al., 2009). Our data demonstrate that the three largest glaciers had low velocities across their tongues: on Ngouzmpa and Pangbung glaciers, velocity remained below 12 ma$^{-1}$ for all 10 bands, whilst on Khumbu

Glacier velocities started high (> 30 ma$^{-1}$) in band 1, but rapidly reduced to below 12 ma$^{-1}$ for the remaining 9 bands (Fig. 7 and 8). We suggest that these slow velocities promoted pond coalescence and growth (Benn et al., 2012, 2017) and also reduced the frequency of pond drainage, by limiting the number of open crevasses (e.g. Miles et al., 2017), and thus increasing pond area.

Generally, the smaller glaciers in the region do not have large, extensive ponding at their termini, but the ponded area

on all seven of the smaller glaciers increased during the study period (Fig. 2). For example, the ponded area on Ama Dablam Glacier increased by 38,958 m$^2$ (48 %) 2015 to 2018 (Fig. 2b). Similar to the larger glaciers, pond number on the seven smaller glaciers did not undergo net change. Thus, given the overall increasing trend in area of the ponds but variations in the changes



to pond number across the region, we suggest that area increases are also primarily due to the coalescing of smaller area ponds (e.g. Ama Dablam Glacier SI. Fig. 3). Whilst pond number generally shows no clear trends, Ama Dablam Glacier is the

exception, and clearly shows an increase in pond number across all study years (Fig. 2c), which suggests area increase on this glacier could be due to the formation of new ponds in addition to pond coalescing, which is consistent with earlier observations (Watson et al., 2016). This coalescing of ponds may enhance melt rates, by increasing the fetch across the pond and increasing the ponded area (Sakai et al., 2009), and potentially lead to proglacial lake development. Ice velocities are generally higher on the smaller glaciers, and more spatially variable, which may limit the opportunity for pond growth and/or encourage crevasses

formation and thus promote drainage (e.g. Immerzeel et al., 2014; Miles et al., 2017; Watson et al., 2017a). A regional difference was noted in 2016, where pond number increased markedly on glaciers to the west, remained relatively similar centrally and decreased on glaciers to the east. For example, pond number increased by 272 on Pangbung Glacier and by 104 on Sumna Glacier, whilst Khumbu, Lhotse and Lhotse Shar glaciers experienced decreases of 58, 91 and 54 respectively. Pond number on Ngozumpa Glacier in the centre of the region decreased by just 33. We speculate that topographic influence by the

ridge that divides east/west (Fig. 1) may result in basin specific microclimates, which may account for the changes noted here. Generally however, pond number decreased across the region and ponded area increased, subject to interannual variations. For 2016-2018, we used imagery from April, but for 2015, we had to use data from December (SI. Table 1), as Sentinel 2A data are only available from November 2015 and we wanted to ensure that our study period continued directly on from that of Watson et al. (2016), to facilitate comparison. Whilst we acknowledge that using imagery from December 2015 could introduce

seasonal differences into our results, our data show that the trend in pond area, and the lack of trend in pond number, are apparent through all years of our study, suggesting that seasonal variations have a limited impact on our results.

### 4.3. Evaluating the use of Sentinel Data for Remote Sensing Studies

Previous studies of supraglacial and proglacial lakes changes in the Everest region have largely used Landsat imagery (resolution = 30 m, 1 pixel = 900 m$^2$;Table 1; e.g. Gardelle et al., 2011; Nie et al., 2013; Zhang et al., 2015). Here we use

Sentinel 2, which is 10 m resolution (1 pixel = 100 m$^2$). Our results demonstrate that ponds < 100 m² (one pixel in Sentinel data) accounted for 3% - 8% and those < 400 m² (four pixels in Sentinel data) comprised 28% - 59% of total ponds found in the region (SI. Fig. 4) (translating to 0.08% - 0.35% and 2.23% - 8.90% for ponded area respectively). Of the total number of ponds identified, between 55% and 86% of ponds were under 900 m$^2$ (i.e. one pixel in Landsat), which equated to between 7.47% and 30.14% of the total ponded area. As such, using Landsat imagery to map pond changes in the region would have

missed the majority of the total number of ponds, and a large proportion of the ponded area. Whilst these ponds are comparatively small in area, including them in assessments is vital, as they inform us about where ponds are nucleating, and hence controls on their formation. These data also indicate locations that may become ponded in the future, and therefore subject to enhanced melt rates, and/or areas that may eventually coalesce with other ponded sections. Furthermore, recent work has demonstrated that Sentinel-2 imagery has a better spectral contrast between debris-cover ice and supraglacial ponds than

Landsat of RapidEye, which affirms its suitability (Watson et al., 2018).



## 4.4 Controls on Pond Formation

### 4.4.1. Glacier Elevation Profile

Previous work suggests that supraglacial ponds usually begin to form on slopes < 10 ° and larger ponds occur where surface gradients are less than 2°, typically found close to the glacier terminus (e.g. Quincey et al., 2007; Bolch et al., 2012). In some

areas, large number of ponds coincided with slopes of 2-4° and pond frequency was highest at the termini of Imja, Lhotse and Sumna glaciers (Figs. 5 and 6). However, contrary to theory, supraglacial ponds were also found in areas with much greater slope gradients (> 10°), both at the termini and further up-glacier (Figs. 5 and 6). For instance, the highest number of ponds on Ama Dablam Glacier (76 %) were located in the high accumulation zone (band 1-2). We also observed changes in the number of ponds after beaks in slope (Fig. 5), which may reflect localised areas of extensional / compressional flow that would

open or close crevasses, and thus facilitate or reduce pond drainage (Benn et al., 2001, 2012; Gulley and Benn, 2007; Röhl, 2008; Thompson et al., 2012; Mertes et al., 2017).

Our data show that the greatest number and area of ponds occurred in the mid-elevation bands (bands 5-6) and not at the termini (bands 8-10) of our study glaciers (Figs. 5-6). We suggest this is because of the inverted mass balance gradient observed on debris-covered glaciers in the Everest region: thick debris at the terminus supresses melt, whereas thinner debris

further up glacier enhances melt (Bolch et al., 2008; Quincey et al., 2009; King et al., 2018). This pattern of mass balance results in near-stagnant ice velocities over much of the tongue and rapid down-wasting at mid-elevations (Quincey et al., 2007, 2009; Nicholson and Benn, 2013; Juen et al., 2014; King et al., 2017), which produces uneven 'ablation topography' (Nicholson and Benn, 2013). Together, this creates ideal conditions for pond formation in mid-elevation areas: the uneven topography allows water to collect; melt rates are at their maximum, due to the comparatively thin debris cover enhancing ice

melt; and low ice velocities inhibit drainage via crevasse opening. In contrast, pond formation may be more limited close to the termini, due to the much thicker debris insulating the underlying ice and thus reducing both the amount of meltwater available to fill ponds and the rate at which ablation topography forms to contain the water. Our data therefore suggest that the optimal conditions for pond formation are now predominantly found at mid-elevations (bands 5 to 6) on glaciers in the Everest region, compared to the termini identified in previous studies (Reynolds, 2000; Bolch et al., 2008; Quincey et al., 2009; Sakai

and Fujita, 2010). This has important implications for total ice loss rates, as ponds strongly enhance glacier melting (Sakai et al., 2000; Benn et al., 2001; Miles et al., 2016; Watson et al., 2016; Mertes et al., 2017), and could thus expand the area of enhanced ice loss further up glacier.

### 4.4.2. Velocity

Ice velocities strongly influence pond size, pond drainage and pond persistence (Miles et al., 2017). Where velocities are low,

little reorganisation of internal water pathways can occur, resulting in ineffective englacial drainage from the glacier surface, which in turn promots surface ponding (Jordan and Stark, 2001; Benn et al., 2012, 2017). As a result, low surface gradients at glacier tongues generally encourage pond formation and low surface velocities facilitate storage, as opposed to drainage



(Reynolds, 2000; Quincey et al., 2007; Watson et al., 2017a). Our results generally support this, as there were more ponds and a greater ponded area closer to the study glacier termini (bands 8-10), where velocities were lower (Figs. 7 and 8). For example, on Sumna, Imja and Lhoste Shar glaciers no ponds were found in bands 1 or 2 where velocity was > 20 ma$^{-1}$, but as velocity decreased ( < 20 ma$^{-1}$) pond numbers began to increase (Fig. 7). However, as with glacier slope, there are exceptions to this relationship. For example, the peak number of ponds corresponds with peak velocities on Lhotse and near-peak velocities on Nupste (Fig. 7). The cause of these velocity patterns is difficult to elucidate, but we suggest that it may relate to the pattern of debris inputs from the surrounding hillslopes (and hence the debris characteristics and distribution), the location of tributary glaciers and/or meltwater inputs from the surrounding slopes. Overall our data indicate that ice velocities influence pond area and number, but that the relationship is far for simple and requires further detailed study.

### 4.4.3. Ice-Cliffs

All ten of the glaciers included in this study had surface ice-cliffs, and the majority of ponds were associated with adjacent ice cliffs (54 %). This follows theory, as ice cliffs melt rates are three to six times greater than that of debris-covered ice (Kirkbride, 1993; Benn et al., 2001; Röhl, 2008; Watson et al., 2016). Thus, ice cliffs can not only initiate pond formation through surface melt (Sakai et al., 2002; Buri et al., 2016a; Watson et al., 2017b) but also facilitate pond expansion in process of thermo-erosional notching (Josberger, 1978; Röhl, 2006). Despite this, our results show that the proportion of ponds without an ice cliff increased substantially during the study period (2015-2018; Fig. 9). For example on Sumna Glacier, ponded area without an adjacent cliff almost doubled during the study, whereas ponded area with a cliff actually decreased, meaning that there were fewer ponds with an ice cliff than without in 2018 (Fig. 9). Overall, our results show that ponds are expanding and that they are usually associated with ice cliffs (Figs. 2 and 9). However, as ponded area increases, the proportion of the ponds with ice cliffs reduces (Fig. 9). We speculate that this may be because pond growth is outstripping ice cliff formation, however given the rapid growth and decay often observed of ice cliffs (e.g. Sakai et al., 2002; Watson et al., 2017b) this is the opposite trend to what would be expected. It could be suggested that higher resolution (< 10 m) imagery is required to identify the full range of ice cliffs on the glacier surface, with smaller cliffs not picked up by Sentinel 2A, however this requires further investigation.

### 4.5. Future Lake Development

### 4.5.1 Applicability of Lake Classification Models

During this study, two of the ten study glaciers progressed to a new stage of the Komori/Robertson (2008; 2012) proglacial lake classification scheme: Ama Dablam and Lhotse Nup, both progressed from Stage 1 to Stage 2. However, Komori (2008) found glaciers in Bhutan took on average 40 years to pass through Stage 1 and 2, and enter into Stage 3, whilst those the in Aoraki/Mt Cook region took 8-30 years for the same transition (Robertson, 2012). Given the short time-frame of this study (4 years), our results demonstrate that proglacial lakes in the Everest region are evolving rapidly and quicker than other regions, both in the Himalaya and globally. This has important implications for hazard assessments in the region, as these rapid changes



require high temporal resolution monitoring to determine potential changes in GLOF risk and could result in hazardous lakes
forming quickly in new locations. Our results therefore highlight the need for frequent monitoring of the hazards posed by
glacier lake growth in the Everest region, particularly given that Nepal (along with Bhutan) is at the greatest economic risk
from GLOFs (Carrivick and Tweed, 2016).

Two of our study glaciers exhibited characteristics of multiple classes of proglacial lake development (Fig. 10).
Specifically, on Pangbung Glacier, ponds were appearing on the lower ablation zone during the study period, which is
indicative of Stage 1, but some ponds began to coalesce (Stage 2) (Table 3). On Lhotse Shar Glacier, we observed pond
coalescing (Stage 2), but also the stable expansion of the proglacial lake (Stage 3). As such, the proglacial lake development
observed in our study region does not fit within the four stage classification system of Komori (2008) and Robertson (2012;
Table 3). Additionally, changes in ponded area, number of ponds and ice-cliffs were observed on all glaciers (Figs. 2 and 9),
but these substantial differences were not accounted for in the current model. For example, both Khumbu and Nuptse glaciers
remained in Stage 2 throughout the study period, but experienced increases in  percentage area of ice-cliffs (45 % and 76 %
respectively; Fig. 9). Ngozumpa Glacier remained in Stage 3, but showed a marked increase in both ponded area (33 %) and
the number of ponds with ice-cliffs (24 %; Fig. 2c and Fig. 9). As such, the current classification model does not account for
many of the observed changes in pond area and characteristics, which could contribute to proglacial lake growth, and does not
account for glaciers that bridge different categories. Thus, on the basis of our observations, we propose a six-stage
categorisation which accounts for the 'in-stage' changes observed, such as appearance of ice-cliffs and marginal pond
expansion (Fig. 11). The inclusion of two additional stages (shown in red), now include 'in-stage' changes observed in this
study, such as marginal expansion, ice-cliff formation and pond drainage. The inclusion of these stages would enable us to
assign our study glaciers to just one stage, making the model more suitable for evaluating and communicating glacier lake
hazard potential.

**4.4.2 Lake Development Trajectories and Outburst Risk**

A key observation of this study was the coalescing of smaller area ponds on the eastern margins of Khumbu Glacier and the
Ngozumpa terminus (e.g. Fig. 3a and b, i and ii). Despite this, both glaciers remained in the same stage of proglacial lake
classification during the period (Stage 3 and Stage 2 respectively). Using our new classification scheme, Khumbu Glacier
would be classified as Stage 3 (compared to currently assigned Stage 2), suggesting that Khumbu Glacier is further along the
progression of proglacial lake formation than previous classifications have indicated. Our data show that Spillway Lake grew
by 40,565 m$^2$ between 2015 and 2018. Previous work showed that Spillway Lake grew by ~ 10% per year (2001 to 2010) to
reach 258,000 m$^2$ in December 2009 (Thompson et al., 2012) but more recently shrank by 8,345 m$^2$ between 2009 and 2015
(Watson et al., 2016). It was suggested that this reduction in area resulted from supraglacial drainage channel evolution and
lowering of the hydrological base level, which paused the lake's expansion (Watson et al., 2016). Given Spillway lake grew
by 40,565 m² to a size of 286,367 m² (December 2015- April 2018), our results indicate the channel has since stabilised,
allowing Spillway Lake to expand up-glacier.





During the 4 years of our study, Imja Lake expanded by 456 m up-glacier and Spillway Lake increased in area by ~ 14% a$^{-1}$. At the same time, we saw large ponds form via coalescing on the margins of Khumbu and Ngozumpa glaciers, and pond coalescence begin on Ama Dablam Glacier (Fig. 3 and SI. Fig. 2a). We also observed an increase ponded area increased on all of our study glaciers (Fig. 2). As such, our results show that increasing volumes of water are being stored on, and in front of, glaciers in the Everest region, and that this trend has accelerated, when our data are compared to previous assessments (Watson et al., 2016). The glaciers in our study region appear to be rapidly moving along the trajectory of proglacial lake formation, and may be doing so quicker than other regions (Komori, 2008; Robertson, 2012). These developments have major implications for downstream GLOF risk, as lake volume and speed of expansion are key factors in the hazard potential of proglacial lakes (Rounce et al., 2017b). As such, there is an urgent need for high resolution monitoring of ice-surface water volumes and proglacial lake development in the Everest Region.

## 5 Conclusions

This study represents the most up-to-date assessment of supraglacial ponds in the Everest region of Nepal. All 10 glaciers demonstrated an overall increase in ponded area over the period 2015 to 2018, ranging from 13.6 % (Sumna Glacier) to 108 % (Lhotse Nup Glacier). Given the short time span of this study, this represents a marked acceleration in the rate of pond growth compared to that presented by Watson et al. (2016) for the period 2000 – 2015, which utilised higher resolution imagery (~ 0.5–2 m). This is a major concern for proglacial lake formation and thus future risk to downstream communities. Our data suggests that increases in ponded area are predominantly through pond expansion and coalescence, rather than new pond formation. This suggests a transition towards larger surface ponds and lakes. Our study shows the importance of using high resolution imagery, as had medium resolution imagery (e.g. Landsat at 30 m resolution) been used, we would have missed the majority of ponds identified in this study (in the order of 55 % to 86 %). Our data show the conditions for pond formation are now predominantly found at mid-elevations on glaciers in the Everest region, suggesting the area where ponds can form may be advancing up-glacier from the debris-covered tongues, allowing ponds to establish over a larger percentage of the glacier surface. Generally, ponded area and number correspond to ice velocities, but there are notable variations in this relationship, which may relate to debris input locations and/or the presence of tributary glaciers. Ice cliffs were found on all 10 study glaciers, but the proportion of ponds with ice cliffs reduced during the study period, suggesting that the rate of ice cliff formation may be slower than pond expansion in this region. Two glaciers (Ama Dablam and Lhotse Nup) progressed to a new stage of proglacial lake development between 2015 and 2018 and this transition occurred much faster than observed elsewhere (e.g. Komori, 2008; Robertson, 2012). Our results suggest that existing classification schemes do not adequately capture proglacial evolution in the study region, so we propose a new six-stage model that encompass observed changes on our study glaciers. Overall, our results demonstrate that the volume of water stored in supraglacial ponds and proglacial lakes in the Everest region is increasing rapidly, and highlights the need for high temporal resolution monitoring of these features, in order to accurately determine the near-future GLOF risk in their downstream catchments.





## 6 Conflict of Interest

The authors declare that the research was conducted in the absence of any commercial or financial relationships that could be construed as a potential conflict of interest.

## 7 Acknowledgements

We acknowledge two freely available datasets used in this study; EarthExplorer from which the Sentintel-2A data and ASTER DEMs were obtained, and the Randolph Glacier Inventory (GLIMS Glacier Database) for the glacier shapefiles. We are also
grateful to Dehecq et al. (2015) for velocity data.

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

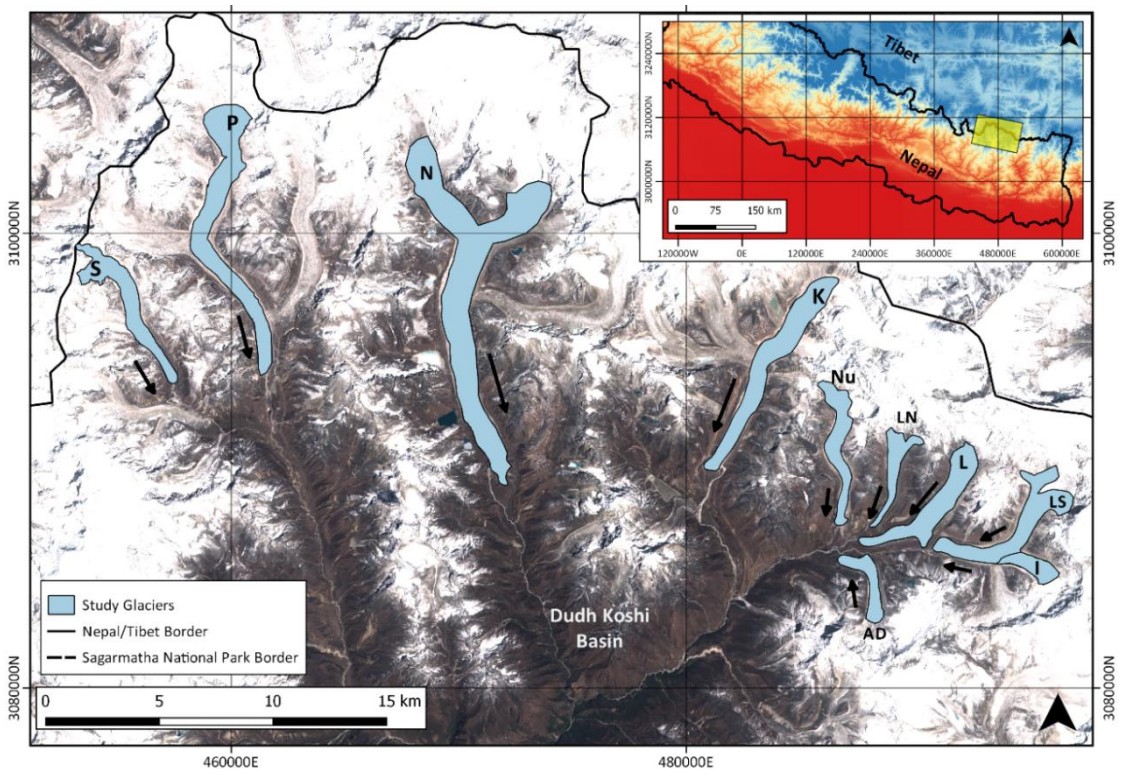

**Figure 1: Location of the 10 study glaciers (blue outline) within the Everest region. Arrows dictate direction of ice flow. Glacier names are as follows: S- Sumna Glacier, P- Pangbung Glacier, N- Ngozumpa Glacier, K- Khumbu Glacier, Nu- Nuptse Glacier, LN-Lhotse Nup Glacier, L- Lhotse Glacier, LS- Lhotse Shar Glacier, I- Imja Glacier, AD- Ama Dablam Glacier. Inset: Location of the**

**Everest Region within Nepal. Background image is surface elevation, derived from ASTER DEM (available at http://earthexplorer.usgs.gov/)).**



| Reference | Date | Imagery | Resolution | Focus |
|---|---|---|---|---|
| **Iwata et al. (2000)** | 1978-1995 | SPOT | Not specified | -Sketch map from SPOT imagery compared to 1987 field survey. |
| **Wessels et al. (2002)** | 2000 | ASTER | 15m | -Water delineated for a single time period. |
| **Bolch et al. (2008)** | 1962-2005 | ASTER Landsat | 15m 30m | -Normalized Difference Water Index (NDWI) and manual delineation used to classify water bodies. |
| **Gardelle et al. (2011)** | 1990-2009 | Landsat | 30m | -Decision tree used to classify lakes using the NDWI. |
| **Salerno et al. (2012)** | 2008 | AVNIR-2 | 10m | -Manual digitalisation of water bodies. |
| **Thompson et al. (2012)** | 1984-2010 | Aerial Photos ASTER Landsat | <1m 15m 30m | -Multi-temporal analysis of Spillway Lake expansion. Glacier area change not reported. |
| **Nie et al. (2013)** | 1990-2010 | Landsat | 30m | -NDWI based classification, but no area changes reported. |
| **Zhang et al. (2015)** | 1990-2010 | Landsat | 30m | -Water bodies manually digitalised but ponded area changes not reported. |
| **Mertes et al. (2016)** | 2014 | GeoEye | <1m | -Conceptual model of supraglacial lake evolution based on Ground Penetrating Radar (GPR) facies analysis of Spillway Lake. |
| **Watson et al. (2016)** | 2000-2015 | GoogleEarth WorldView 1&2, GeoEye, QuickBird-2 | 2m 0.5-0.6m | -Water bodies semi-automatically or manually digitised. Area changes are quantified. |

**Table 1: Previous remote sensing studies of supraglacial water storage in the Everest region (After Watson et al., 2016).**









| Glacier | Area (km²) | Length (km) | Min-Max Elevation (range) (m) |
|---|---|---|---|
| Ama Dablam | 2.13 | 4.40 | 4769 – 5084 (315) |
| Imja | 1.08 | 2.46 | 5023 – 5187 (164) |
| Khumbu | 6.64 | 10.82 | 4956 – 5246 (290) |
| Lhotse | 5.74 | 6.69 | 4715 – 5245 (530) |
| Lhotse Nup | 1.38 | 3.82 | 4954 – 5310 (356) |
| Lhotse Shar | 3.00 | 4.03 | 5008 – 5429 (421) |
| Ngozumpa | 15.10 | 15.76 | 4868 – 5541 (673) |
| Nuptse | 3.06 | 6.20 | 4662 – 5354 (692) |
| Pangbung* | 11.34 | 13.14 | 4742 – 5380 (638) |
| Sumna* | 5.34 | 7.39 | 4888 – 5502 (614) |

**Table 2: Characteristics of the 10 study glaciers, showing area, length, elevation and direction of flow. *Indicates glaciers not**
**included in the Watson et al. (2016) study.**





**Figure 2: Changes in supraglacial ponds observed on all 10 glaciers March 2015- April 2018; (a) ponded area as a percentage of total glacier area, (b) total number of ponds and (c) total ponded area. Bars are colour coded by date. AD-Ama Dablam Glacier, I-Imja Glacier, K-Khumbu Glacier, N-Ngozumpa Glacier, NU-Nuptse Glacier, L-Lhotse Glacier, LN-Lhotse Nup Glacier, LS-Lhotse Shar Glacier, P-Pangbung Glacier, S-Sumna Glacier.**












**Figure 3: Ponded area change for the Ngozumpa (a) Khumbu (b) Pangbung (c) and Sumna (d) Glaciers, for the period 2015-2018. Inset (a) up-glacier expansion of Spillway Lake, inset (b) pond expansion along the lower eastern margins of Khumbu Glacier, inset (c) pond expansion on the glacier tongue, (d) pond proliferation on the lower ablation zone 2015-2017 and drainage along the western** 690 **margins 2017-2018. Arrows indicate direction of ice flow. Background image is 2018 Sentinel-2 imagery (from USGS at** *http://earthexplorer.usgs.gov/***).**

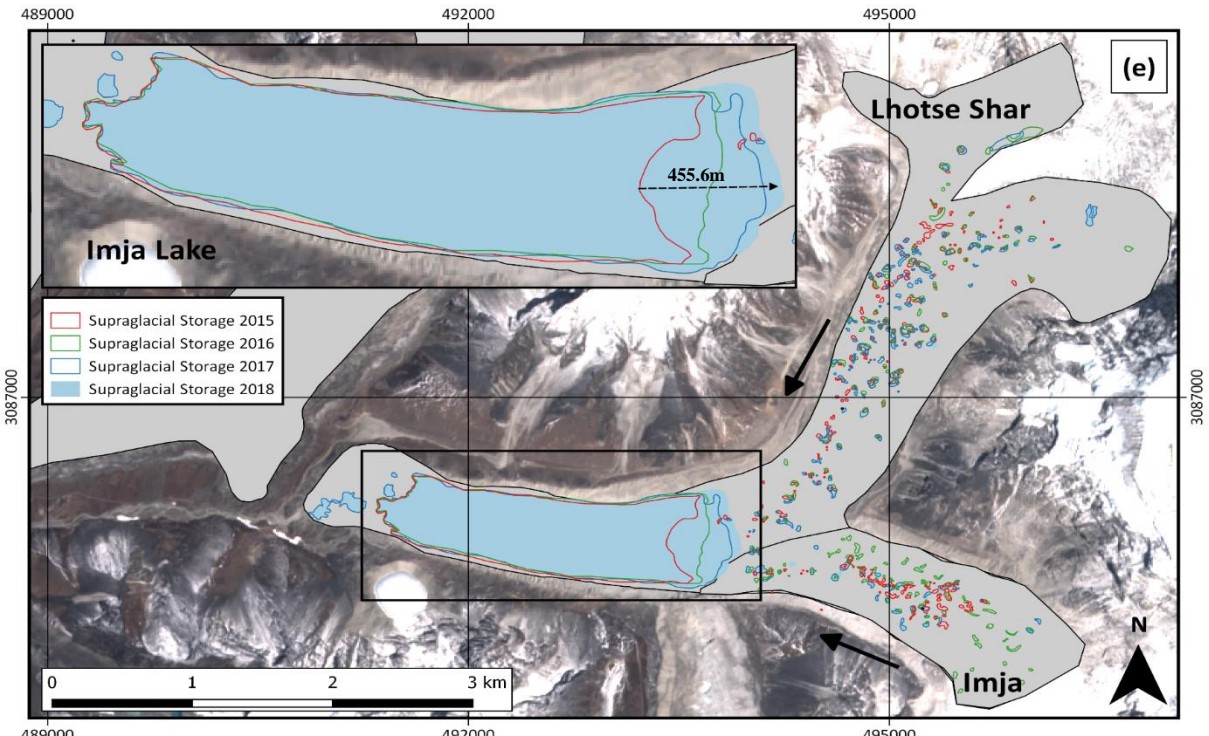

**Figure 4: Area change of Imja Lake, in the southeast of the region for the period 2015-2018. Arrows indicate direction of ice flow. Background image is 2018 Sentinel-2 imagery (freely available from USGS at http://earthexplorer.usgs.gov/).**



**Figure 5: Number of ponds found on the glacier surface April 2018 compared to glacier elevation/slope at 10% glacier area distance bands (0-10 = band 1, 10-20= band 2 etc.) Dashed red line indicates breaks of slope.**






**Figure 6: Ponded Area found on the glacier surface April 2018 compared to glacier elevation/slope at 10% glacier area distance bands (0-10 = band 1, 10-20= band 2 etc.)**




**Figure 7:  Number of ponds found on the glacier surface April 2018 compared to glacier velocity at 10% glacier area distance bands (0-10 = band 1, 10-20 = band 2 etc.**



**Figure 8: Ponded Area found on the glacier surface April 2018 compared to glacier velocity at 10% glacier area distance bands (0-10 = band 1, 10-20 = band 2 etc.)**




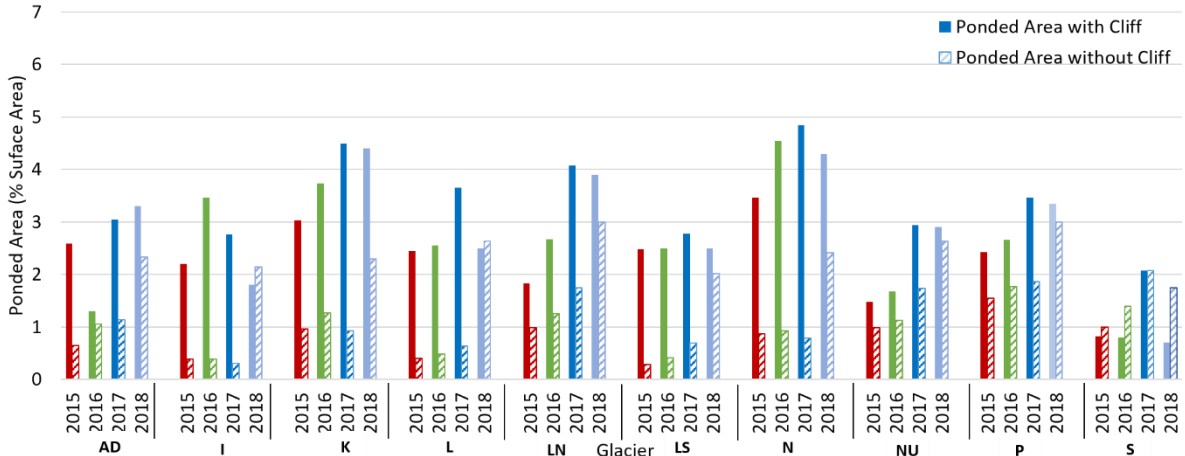

**Figure 9: Percentage of ponds with and without ice-cliffs, displayed as percentage of total glacier surface area. AD- Ama Dablam Glacier, I- Imja Glacier, K- Khumbu Glacier, N- Ngozumpa Glacier, NU- Nuptse Glacier, L- Lhotse Glacier, LN- Lhotse Nup Glacier, LS- Lhotse Shar Glacier, P- Pangbung Glacier, S- Sumna Glacier.**




| Stage of Lake Development | Description |
|:---:|:---|
| 0 | No supraglacial ponds. |
| 1 | Appearance and growth of supraglacial ponds in the lower ablation zones |
| 2 | Coalescing of supraglacial ponds to form larger, ice dammed ponds. |
| 3 | Stable expansion of ponds up-glacier and shift to moraine dammed proglacial lake. |
| 4 | Glacier retreats out of the proglacial lake. |



**Table 3: Stage of lake development following Komori (2008) and Robertson (2012).**



**Figure 10: Stage of lake development change observed over the study period, from 2015 (bottom) to 2018 (top). AD- Ama Dablam Glacier, I- Imja Glacier, K- Khumbu Glacier, N- Ngozumpa Glacier, NU- Nuptse Glacier, L- Lhotse Glacier, LN- Lhotse Nup Glacier, LS- Lhotse Shar Glacier, P- Pangbung Glacier, S- Sumna Glacier. Background image is 2018 Sentinel-2 image (available from USGS at http://earthexplorer.usgs.gov/).**



| Stage 1 | Stage 2 | Stage 3 |
|---|---|---|
| Appearance and growth of supraglacial ponds in glacier lower ablation zones. | Increased pond frequency up-glacier and formation of ice cliffs. | Pond expansion and/or pond drainage. 765 |
| **Stage 4** | **Stage 5** | **Stage 6** |
| Coalescing of supraglacial ponds to form larger terminal lakes. | Characterised by the stable expansion of lakes up-glacier. | Glacier retreats out of the proglacial lake. 770 |

**Figure 11: Updated Komori (2008) and Robertson (2012) conceptual lake classification model, to include 2 additional stages**
775 **accounting for 'in-stage' changes observed in this study; Stage 2: recognises the role of ice-cliffs and changes in surface debris leading to pond formation and decay; Stage 3: gives marginal based expansion of ponds and pond drainage events their own category (After Komori, 2008; Robertson, 2012; Mertes et al., 2016).**