# Peer review of "Supraglacial pond evolution in the Everest region, central Himalaya, 2015-2018."

_The Cryosphere, 2019_

## Referee Comment (RC1) · C. Scott Watson (Referee) · 19 Mar 2019

Taylor and Carr present an assessment of supraglacial pond evolution in the Everest region of Nepal over a ~2.5 year period using Sentinel-2 imagery. As they detail, the region has seen extensive research investigating glacier hydrology and supraglacial pond dynamics. For this reason, I would expect the paper to offer new insight into the changes in this area, rather than a simple extension of previous studies by a few years. They suggest the need for a new method of ranking glaciers based on their stage of lake development (which a novel aspect of the paper) but this is not adequately supported by the results. A spatio-temporal assessment of pond evolution needs to consider the change at specific ponds/pond basins, e.g. the recent work by Benn et al. (2017). Similarly, grouping pond change into elevation bins is problematic in the study region because of the low gradient nature of the glaciers. Other studies have opted for distance based metrics e.g. Thompson et al. (2016). Nonetheless, it is not clear which DEM (or the date) was used in the paper for the elevation groupings. Objectives 2 and 3 have been addressed in other studies, so again there are limited insights here. The spatial resolution of the imagery used (10 m/ 100 $m^2$) does not allow the authors to address how many ponds were smaller than 100 $m^2$ for their Objective 2.

My main concern is the lack of detail and errors in the methodology e.g. lack of an uncertainty assessment, and justifying and specifying the use of a thermal band for pond classification, when Sentinel-2 does not have a thermal band, and nor would it be suitable for pond classification. This naturally leads to questioning the robustness of the results over such a short timeframe when previous studies have shown large intra- and inter-annual variation in pond coverage. To highlight this, I have shown the pond data for 2016 and 2017 on Khumbu Glacier from Watson et al. (2018), which was derived from 0.5 m and 3 m resolution imagery respectively. In both cases, the total pond area reported by Taylor and Carr for the same years is approximately double that of Watson et al. (2018), so further investigation is required. There are Taylor and Carr classified ponds that I see no evidence of in the raw imagery (I have provided an example on Khumbu Glacier). There are also issues in the discussion, where the authors refer to ponds in the 'high accumulation zone' despite glacier accumulation zones not appearing in the study.

Conclusions of rapid pond expansion discussed in the context of outburst flood hazards is a sensitive issue in a region where $7 million was recently spent on glacial lake hazard mitigation work. Therefore, I believe it would be unfortunate (and potentially problematic) for residents to be unnecessarily alarmed or misled based on these results. The authors should improve their methodology and demonstrate that the results are robust. For the reasons outlined above and detailed below, I cannot recommend publication in The Cryosphere and I have rejected the paper on this basis.

I have detailed specific problems below and I hope the authors would use this to help prepare a substantially revised study, which I would gladly read. The authors could include comparisons to the reference datasets of Watson et al. (2018) in their study, which can be provided on request. The authors should also ensure that the results are communicated in a way that considers the sensitive nature of glacier hazards.

Kind regards,

C. Scott Watson

**General comments**

Objectives:

- Objective 1: this is difficult to address over such a short time scale in your study (December 2015 to April 2018). Recent work has shown that in some cases pond coverage is expanding in the region; however, the takeaway message from Watson et al. (2016) was that pond coverage was expanding in

some locations but there was large annual and inter-annual variation. Seasonal pond expansion was especially large. Therefore making conclusions based on less than three years of data is problematic.

- Objective 2: 100 m$^2$ is the pixel area of Sentinel-2 imagery so you cannot state what proportion of ponds were less than this without subpixel analysis or use of validation data. Nonetheless, the issue is already addressed in detail by other studies e.g. Salerno et al. (2012), Miles et al. (2016), Watson et al. (2016) and Watson et al. (2018). Your results are not discussed in the context of these studies. Without using subpixel analysis or a validation dataset, you cannot perform this objective.
- Objective 3: this was recently addressed by Salerno et al. (2017) and Watson et al. (2017a) for these specific glaciers. Your findings should be discussed in the context of previous studies.

Methods:

- There is no uncertainty assessment in the pond or cliff classification so it's not clear what is statistically significant and what isn't. This could be carried out using commonly used +/- one or half pixel uncertainties in the classification.
- There is no information regarding how the Maximum Likelihood Classification (MLC) was carried out (number of training sites, distribution, validation etc).
- The authors state they used bands 5 and 7 but no justification is given. They state they used band 7, which is incorrectly referred to as a thermal band, because 'thermal wavelengths are absorbed by water bodies so their addition aided the classification substantially'. References should be provided. Nonetheless, this is concerning because Sentinel-2 does not have a thermal band.
- The single reference used to support the Maximum Likelihood Classification '(Tiwari et al., 2016)', specifically state that they did not use the method for classifying water because there were 'no prominent water bodies' in their study. The conference paper was focused on classifying debris cover from clean ice.
- It is not clear how many ponds were manually classified vs classified using MLC. Figures 3 and 4 show a clear difference in the appearance of pond outlines, with many appearing very smooth and rounded but few displaying edges suggestive of a pixel classification. There are also many ponds that have been classified outside of the glacier masks.
- Please provide detail about how/if the frozen or partially thawing nature of the pond surfaces in April would affect the classification. Provide examples of the underlying imagery used. Currently all examples are hidden by the glacier masks or the pond polygons. A separate figure should be added showing the classification procedure. The images 2016-2018 are progressively later in April, could this lead to larger ponds in the later images?
- Please provide detail on the ice cliff digitization, perhaps on the same figure as the pond classification.
- Please provide detail on the elevation bands. I'm not clear how they were derived or what they correspond to. Perhaps add a figure in the supplement.
- Glacier names: I can't find reference to Pangbung and Sumna Glaciers. International Centre for Integrated Mountain Development and other publications refer to them as Bhote Kosi Glacier and Melung Glacier.

Results:

For comparison, I have taken the published data of Watson et al. (2018) for Khumbu Glacier in 2016 and 2017. The reference datasets were derived from 0.5 m resolution Pleaides imagery (Nov 2016) and from 3 m resolution PlanetScope imagery (Nov 2017). These are not directly comparable with the April data of Taylor and Carr, but nonetheless should have a similar pond area since both datasets are outside the melt season. November 2016 should also be comparable to April 2017, since ponds may have shrunk over the winter, but not expanded. I have also included the pond area delineated using a NDWI classification applied to Sentinel-2

imagery (Nov 2016 and 2017). The reference dataset was also derived for other glaciers in your study, so consider requesting this from the authors or collaborating with one of the several research teams that has data in this region. The classification of Taylor and Carr Khumbu Glacier contains approximately double the ponded area compared to that of Watson et al. (2018). In the case of two instrumented ponds on Khumbu Glacier Watson et al. (2017b), water levels were observed to be rising in mid-late April (see their Fig. 6), so the difference between pre- and post-monsoon water levels requires investigation.

| | Watson et al. (2018) (November) Reference dataset | Watson et al. (2018) (November) Optimised Sentinel-2 classification | Taylor and Carr (April) |
|---|---|---|---|
| Glacier (date) | Pond area (m$^2$) (number) | Pond area (m$^2$) (number) | Pond area (m$^2$) (number) |
| Khumbu (2016) | 195,100 ± 17,000 (287) | 197,700 (188) | 369,168 (165) |
| Khumbu (2017) | 191,500 ± 76,000 (211) | 180,300 (172) | 427,386 (225) |

Missing ponds:

I cannot distinguish between the 2017 (blue outlines) and 2018 (blue fill) on Khumbu Glacier (Figure 3). Nontheless, I have taken a distinct area of ponds and compared the Taylor and Carr classification with the Sentinel-2 imagery they used in the study (I have screenshotted band 8, 4, 3 false colour composites). I cannot see evidence of the yellow highlighted ponds in either of the two Sentinel-2 images that the classification should be based on (2017 or 2018).

Taylor and Carr Figure 3 classification overlaid on Sentinel-2 imagery:

[Figure]

18 April 2017:

[Figure]

23 April 2018:

[Figure]

With transparency:

[Figure]

**Specific comments**

L8. 'which can potentially represent a hazard'

L12. Is this the same chain identified by Watson et al. (2016)?

L15. Clarify 'general theory'

L16. Does this affirm the conclusions of Watson et al. (2018)?

L42-43. The Watson et al. 2016 ref doesn't belong here.

L55. 'and presence of ice cliffs'

L96-97. 'high' is not specific.

L110. In the abstract you state the resolution is 10 m. Here you state <10 m.

L117 'minimize' to be consistent with American English. Check for consistency throughout.

L118. Specify the two tiles used in the mosaic.

L120. Use 'non-glacier' instead of 'land'?

L120-121. There are clean ice areas on Khumbu and Ngozumpa Glaciers within your debris mask. Add the RGI citation.

L124. 'semi-automatically'

L125. How many training sites? Distributed how? Did you aim to just include pure pond pixels or mixed pixels also?

L126-128. Band 5 is not the NIR band, nor does 7 have a thermal wavelength. Sentinel-2 does not have a thermal band.

L136. Do you mean the ASTER GDEM, or an ASTER DEM product (e.g. AST14DEM)? Specify the citation and date for whichever was used.

L139. What is the uncertainty in this velocity product? On Fig.7 you show velocities of 2 metres per year down to less than 0.5 metres per year. Detail how this can be resolved in a 120 m spatial resolution product.

L140. Specify what type of bands. Elevation, distance?

L141. Specify how the cliffs were mapped. As polygons e.g. Herreid (2018)? Or lines e.g. Watson et al. (2017a). The former requires fine resolution imagery/ DEM so it's not clear how this was done on Sentinel-2 imagery, especially when resolving steeply-sloping cliffs. What was the uncertainty in the delineation?

L157. 'markedly'

L174. Remove the decimal point. 456 m

L175. Remove the decimal places from '1,493,142.68'

L191-193. I'm not sure why this comparison is made. As a proportion of the glacierised area, pond coverage on Lhotse Shar and Imja were similar (Fig. 9) and you show the area of Lhotse Shar to be approximately three times that of Imja.

L204. How was the average slope calculated and does it related to the full study glacier (including clean ice), or the smaller mask that you are using?

L207-209. Clarify what you mean by '10 equal elevation bands' in section 2.4 and how this varies glacier to glacier. Consider a methods figure showing the process.

L217-218. Accumulation zones are not part of your glacier mask so this is incorrect.

L221. What do you mean by 'higher frequency ponding'? Is this the same as 'numbers of ponds' used in the previous sentences. If so perhaps change for consistency.

L222. Define what you are calling a 'break in slope'. Some glaciers e.g. Nuptse and Khumbu have similar 'steps' in the elevation profile in Fig.5 but no labeled break in slope. Does this relate to local slope, or the glacier gradient?

L234. Add a space between units here and throughout and be consistent with the figure axis label.

L246-247. The percentage change is actually quite small, so could this be within levels of uncertainty?

L245. Be consistent with the number of decimal places.

L247. Remove space. '3.3 %'. Check throughout.

L248. Remove '-' in ice-cliff and be consistent throughout.

L249. Do you mean they covered an additional 1.6% of the glacier surface area?

L254-255. This is text for the discussion.

L259. You need to give the reader some information in the text regarding what these stages mean. 'ice-dammed' what?

L260. Space between units.

L263. Fig. 10b does not detail the ablation zone. 'Ablation zone' and 'accumulation zone' need to be specifically detailed in the context of the study and figures in order to discuss them.

L278-281. There is an assumption made here that Lhotse Glacier is going to develop a proglacial lake, so you need to detail why. There is a small village (Chukhung) nearby, and they would likely be very concerned to hear these results.

L283. Could precipitation have caused the pond expansion trend observed? i.e. could 2018 have seen more snowfall than 2017 or 2016?

L292-294. How prevalent are crevasses on the study glaciers?

L296. Provide a reference here. I think (Benn et al., 2012).

L301. This may be true, but the reader cannot tell unless you present an analysis of how individual ponds have changed.

L307. But only if the expanded pond area is suitably aligned.

L311. You are 'suggesting' things that have already been detailed in the references. The reader wants to know what has changed. What is your study showing that's new? Are these velocities different from previous studies, or does your data demonstrate the same thing and reaffirm those studies?

L321. 'could be due to the formation of new ponds in addition to pond coalescing' is not specific. What do your specific data show?

L323. Where do you show that ice velocities are higher on the smaller glaciers? This is not consistent with my understanding.

L330. What about weather, which is more relevant to your study time scale?

L334. I'm not sure you've made a specific comparison in terms of pond area/number?

L336. This does not suggest seasonal variations have limited impact. Your data do not show this. It suggests that you expect the trend you found to exceed any seasonal variation.

L338-350. This is problematic because there is an assumption that Sentinel-2 can observe the full size distribution of ponds. Salerno et al. (2012), Watson et al. (2016) and (2018) deal with the issue of pond omissions/ commissions using satellite imagery of variable resolution and include even finer-resolution imagery as ground truth (~ 3 m or less). Therefore, I don't think this section adds anything new. Nonetheless, this paragraph should refer to these previous studies.

L340. 100 m$^2$ is the pixel area of Sentinel-2 imagery so you cannot state what proportion of ponds were less than this without subpixel analysis or use of validation data.

L342. There is no supplementary Figure 4.

L350. Watson et al. (2018) also provided a pond-by-pond comparison of the role of fine resolution vs coarser resolution imagery.

L355. 'large number of ponds coincided with slopes of 2-4°'. This analysis is not presented. Is it based on the ASTER DEM?

L357. Slope or gradient. Terminology needs to be clarified since local slope, and glacier-scale slope/ gradient, have very different meanings and roles.

L363. 'We suggest' should be referring to a new interpretation, not something widely known. 'Affirm' or 'confirm' should be used when your results support already published works. This should be changed throughout the paper.

L373-375. How does this compare with Watson et al. (2016)? Has the spatial distribution of ponds changed significantly?

L381. 'promotes'

L395. The correct Watson reference for ice cliff melt is Watson et al. (2017c).

L404. Watson et al. (2017a) found cliffs 20-40 m in top edge length to be most prevalent in the region, which indeed suggests that Sentinel-2 can observe some of the larger cliffs, but the uncertainty will be large. The main issue is that the surface of steeply sloping cliff topography is not resolved unless using a 3D surface.

L414. Your study was less than 4 years. December 2015 to April 2018.

L449. Why are ponds classified outside the mask on Supplementary Figure 2?

L485. Specify the ESA usage statement/citation for Senitnel-2 data.

Figure 1. Can you add the rest of the glacier outlines and specify debris vs clean ice. Specify UTM grid zone 45N. Suggesting changing the inset to lat lon.

Figure 2. Incorrect units (km$^2$).

Figure 3. There are some ponds outside the glacier outline. Are these included in the calculations? I can't see a clear different between 2017 and 2018 outlines in some cases (e.g. on Khumbu Glacier).

Figures 5-8. Graph scales vary between glaciers, which makes visual comparisons difficult.

Figure 6. 'glacier elevation/slope at 10% glacier area distance bands'. These need detailing in the study and you should show them on Figure 1.

Figure 7. Change the velocity label to 'm yr$^{-1}$' or 'm a$^{-1}$' with superscript,

Figure 10. There appears to be some issue with the pond outlines on the lower panel. I suspect glaciers west of Nuptse have pond outlines (with a line thickness), whereas those west of Nuptse have a filled polygon but no outline, making the ponds look less prominent. Check and correct if required. Label the panels. Did this influence your interpretation of the potential for catchment microclimates?

Figure 11. Very similar to Table 3. Condense to a table or Figure.

Supplement:

Figure 1. Units are incorrect.

Figure 2. State the imagery date.

Table 2. Specify the full imagery date.

References

Benn, D.I. Bolch, T. Hands, K. Gulley, J. Luckman, A. Nicholson, L.I. Quincey, D. Thompson, S. Toumi, R. and Wiseman, S. 2012. Response of debris-covered glaciers in the Mount Everest region to recent warming, and implications for outburst flood hazards. *Earth-Science Reviews.* **114**(1–2), 156-174. http://dx.doi.org/10.1016/j.earscirev.2012.03.008.

Benn, D.I. Thompson, S. Gulley, J. Mertes, J. Luckman, A. and Nicholson, L. 2017. Structure and evolution of the drainage system of a Himalayan debris-covered glacier, and its relationship with patterns of mass loss. *The Cryosphere Discuss.* **2017**, 1-43. 10.5194/tc-2017-29.

Herreid, S. and Pellicciotti, F. 2018. Automated detection of ice cliffs within supraglacial debris cover. *The Cryosphere.* **12**(5), 1811-1829. 10.5194/tc-12-1811-2018.

Miles, E.S. Willis, I.C. Arnold, N.S. Steiner, J. and Pellicciotti, F. 2016. Spatial, seasonal and interannual variability of supraglacial ponds in the Langtang Valley of Nepal, 1999–2013. *Journal of Glaciology.* **63**(237), 1-18. http://dx.doi.org/10.1017/jog.2016.120.

Salerno, F. Thakuri, S. D'Agata, C. Smiraglia, C. Manfredi, E.C. Viviano, G. and Tartari, G. 2012. Glacial lake distribution in the Mount Everest region: Uncertainty of measurement and conditions of formation. *Global and Planetary Change.* **92-93**, 30-39. http://dx.doi.org/10.1016/j.gloplacha.2012.04.001.

Salerno, F. Thakuri, S. Tartari, G. Nuimura, T. Sunako, S. Sakai, A. and Fujita, K. 2017. Debris-covered glacier anomaly? Morphological factors controlling changes in the mass balance, surface area, terminus position, and snow line altitude of Himalayan glaciers. *Earth and Planetary Science Letters.* **471**, 19-31. https://doi.org/10.1016/j.epsl.2017.04.039.

Thompson, S. Benn, D. Mertes, J. and Luckman, A. 2016. Stagnation and mass loss on a Himalayan debris-covered glacier: processes, patterns and rates. *Journal of Glaciology.* **62**(233), 467-485. http://dx.doi.org/10.1017/jog.2016.37.

Watson, C.S. King, O. Miles, E.S. and Quincey, D.J. 2018. Optimising NDWI supraglacial pond classification on Himalayan debris-covered glaciers. *Remote Sensing of Environment.* **217**, 414-425. https://doi.org/10.1016/j.rse.2018.08.020.

Watson, C.S. Quincey, D.J. Carrivick, J.L. and Smith, M.W. 2016. The dynamics of supraglacial ponds in the Everest region, central Himalaya. *Global and Planetary Change.* **142**, 14-27. http://dx.doi.org/10.1016/j.gloplacha.2016.04.008.

Watson, C.S. Quincey, D.J. Carrivick, J.L. and Smith, M.W. 2017a. Ice cliff dynamics in the Everest region of the Central Himalaya. *Geomorphology.* **278**, 238-251. http://dx.doi.org/10.1016/j.geomorph.2016.11.017.

Watson, C.S. Quincey, D.J. Carrivick, J.L. Smith, M.W. Rowan, A.V. and Richardson, R. 2017b. Heterogeneous water storage and thermal regime of supraglacial ponds on debris-covered glaciers. *Earth Surface Processes and Landforms.* 229-241. http://dx.doi.org/10.1002/esp.4236.

Watson, C.S. Quincey, D.J. Smith, M.W. Carrivick, J.L. Rowan, A.V. and James, M. 2017c. Quantifying ice cliff evolution with multi-temporal point clouds on the debris-covered Khumbu Glacier, Nepal. *Journal of Glaciology.* **63**, 823-837. http://dx.doi.org/10.1017/jog.2017.47.

---

## Referee Comment (RC2) · Anonymous Referee #2 · 1 May 2019

Review of 'Supraglacial pond evolution in the Everest region, central Himalaya, 2015-2018', by C Taylor and R Carr

The analyses by Taylor and Carr have used Sentinel-2 multispectral data for 2015 to 2018 to identify supraglacial ponds on 10 glaciers in the Everest region of Nepal. The study thus seeks to extend previous analyses of pond incidence by 4 years, and to evaluate the state of glacier lake development in the region.

The analysis has several major errors that need to be reconsidered before a review can be completed, but more unfortunately, the framework for the study does not look to provide any new insights into the prevalence of supraglacial ponding, nor for the development of supraglacial lakes in the region. The current results are impossible (reported supraglacial ponded area several times larger than glacier areas) and over-interpreted in the discussion. I recommend that the manuscript be rejected and returned to the authors.

The following major points deserve consideration: 1. Lack of clear motivation and value for the study. The study aims to use Sentinel 2 data to extend the analysis of Watson et al (2016) to present, but provides very little justification for this aim or for the approach used. Why will 4 years of additional annual data be helpful in assessing glacier lake development? Glacial lakes usually take 10-30 years to develop (see the Imja lake development history, for example). Can you be sure that the changes over a 4-year period are not simply due to noise – past studies have identified high interannual and seasonal variability for these features, and a time series length of 4 gives low confidence. Altogether there seems to have been little critical evaluation of the research question and suitability of the analysis to achieve an answer.

2. Key references and understanding missing. The manuscript exhibits a low-level of understanding of past work on supraglacial ponds and ice cliffs, and their association and co-evolution. The use of terms like 'slope gradient' convey missing basic understanding, and in several instances the results of other studies are misinterpreted. The list of past efforts to map supraglacial ponds is incomplete, and only one alternative method to map ponds is even mentioned, with no consideration for its use. The classification scheme for glacial lake development is fundamentally misunderstood as well, leading to the development of the authors' own classification scheme. Fundamentally, the 'glacier' outlines used in the figure, and the text within the manuscript, indicate that the authors have a misunderstanding of what a glacier is – they have delineated only the debris-covered areas, but discuss accumulation areas within these bounds.

3. Incomplete methodological description. There are many key details missing with respect to the pond classification method (whether atmospheric corrections were performed, how many samples were used for each landcover class, which landcover
classes were used, even the physical basis of the scheme and how MLC works), for the slope analysis (why surface slope is a predictor for ponds, whether slope is the same as gradient, and any details for the slope calculation itself), etc. In addition, the velocity data analysed in the study were derived and provided by Dehecq et al (2015) but repeatedly presented as 'our' velocity data. At the same time, virtually no details are provided on its derivation (even what sensor it is derived from).

4. No consideration of pond seasonality or interannual variability. The 4 years of data are erroneously interpreted to directly imply a trend, despite the low sample size and low study duration. In addition, the analysed scenes correspond to an unusual period of the year for analysis (April) when cloud cover can be very problematic – no discussion is included for cloud identification and removal, which can be particularly problematic for pond delineation. The April scenes are also susceptible to the stochastic filling of small basins as winter and spring snow melts, which can lead to additional noise in the pond-cover time series. Worse, the study has also included a December scene, when ponds are often frozen-over and covered by snow, which also requires further consideration and discussion.

5. Erroneous/impossible results. The reported total pond areas are on the order of many km2, whereas the glaciers themselves are only a few km2 each. This may be a basic calculation mistake, but it is concerning that the proofreading did not pick up on this mistake. As someone familiar with the glaciers in question, the 2018 data look suspicious, indicating a higher pond coverage that I would have expected. This could be related to key point 4.

6. No advance in understanding based on the study. The addition of 4 years of pond coverage itself provides little value, as the authors have not considered seasonality (a task which modern optical sensors provide opportunity for). The authors have tried to reinterpret the stages of glacial lake development, but I have issues with their updated framework, which also does not seem to be based on the results within the manuscript. Most specifically, there is no evidence that cliffs or ponds increase in incidence up-
glacier with time or that ponds pre-date cliffs (suggested stage 2). Rather, the earliest observations of glacier surface morphology in the area (e.g. Watanabe et al, 1986) report debris, cliffs, and ponds with a similar distribution to that found today, with the change that the entire debris area has expanded slightly up-glacier. In addition, pond expansion and drainage is an ongoing, circular process that happens on seasonal and multi-year timescales (e.g. Benn et al, 2017; Miles et al 2017; Miles et al, 2017b). Fundamentally, perched ponds will eventually drain if they exist at all (Mertes et al, 2016) so it is unclear how this represents a distinct stage of lake development.

Detailed comments through results; I feel that there are enough errors that need to be amended that feedback beyond results is not yet constructive.

L8. Proglacial lakes can represent a major hazard to downstream communities, but do not always. In this instance I think you mean moraine-dammed lakes, or ice-marginal glacial lakes. See Carrivick and Tweed, 2013.

L8. Why only S2A, and not also S2B? Not that the 10m spatial resolution only applies to some bands.

L27. Dehecq et al 2018 (should be 2019) is an odd choice of reference here, as that study did not look at impacts on communities.

L31-35. See Harrison et al, 2018, which discusses the historical incidence and future development of glacial lakes.

L40. -0.22+/-0.12 m w.e./a is not 'strongly negative'; this is similar to the global historical average (e.g. Haeberli et al 1999).

L46. I would suggest qualifying this statement: 'are often characterised by...' as there is considerable variability across the region, and not all of the glaciers follow this description.

L63-78. I think there may be some referencing confusion here. Miles et al, 2016 (in Annals of Glaciology) did discuss these points, but you have referenced one of several

TCD
Miles et al 2017 publications (in Journal of Glaciology).

L73-75. This is backwards to me; according to the Brun, Buri, and Watson studies, the ponds enhance the ice cliff melt (not cliffs enhancing the pond melt) and certainly are the stimulus for calving.

L80. Watson et al, 2016 should be in this list.

L81. This point should be attributed to Salerno et al (2012) and Watson et al (2016), which bot examined these same glaciers with high-resolution imagery to quantify the areas missed by coarser sensors.

L86-91. Why is the addition of 3 years of coverage a scientific priority? There is little clear motivation for this analysis, especially as the study has not integrated the results from Watson et al (2016) to provide a continuous perspective from 2000 to present. Of the stated objectives, (1), (2) and (3) have already been carried out by previous investigators, so it is important to clarify how your analysis will differ from those of Gardelle, Salerno, and Watson, respectively.

L96. Presumably this should be 'high-elevation accumulation areas'

L101. Why the reference for Immerzeel et al (2010) here? That study did not focus on the Everest region, nor did it examine GLOFs at all.

L103-108. Thank you for formalising the pond/lake name convention.

L110. Not all S2 bands are at 10m resolution, but also, none are

did you mitigate this, which could cause confusion for your algorithms (and for the comparability between years)?

L125. Please provide the full set of training classes, as well as the number of training sites used for each class, and a comparison of the spectral characteristics derived from each class (as in Gardelle et al, 2011).

L127. For consistency, please specify the wavelengths of these bands.

L128-133. This is a very cursory overview of glacial pond/lake mapping techniques, with no discussion of spectral methods based on the normalised difference water index (Watson et al 2018) or other indices (Gardelle et al, 2011) or even the physical basis for delineating water using wavelengths beyond visible light.

L136. More details are needed with respect to your derivation of a slope map. What algorithm did you use? Did you simply calculate slope at each pixel based on its neighbours, or did you perform a more sophisticated analysis (Quincey et al 2007; Miles et al 2017 JGlac; King et al 2018)? Also, the terms 'slope' and 'gradient' mean different things, and 'slope gradient' is literally the spatial rate of change of slope.

L141. More discussion is needed with regards to the ice cliff mapping. Did you simply mark cliff edges (i.e. Thompson et al 2016), or map cliff outlines (Brun et al 2018), and how did you assess your error in this regard? Most studies that map ice cliffs use even higher resolution imagery as cliffs can have narrow planimetric areas, so how did this work practically? How did you associate cliffs and ponds to one another?

L145-157. None of these ponded area values are plausible. All are greater than the area of the individual glaciers by at least an order of magnitude.

L158-169. I find it quite difficult to accept the perceived increase in supraglacial ponded area over a 4-year period, given the potentially-large interannual variability of supraglacial ponds (Miles et al 2017, JGlac).

L187-189. You have given 2 examples of coalescence at large lakes resulting in areal

TCD
increase, but this statement needs a specific analysis – there are >6000 ponds in your study, so it is difficult to consider that this is 'predominantly' the mechanism of expansion. How many lakes drained over the study period, and how many new ponds formed over the same years?

L196-197. A comparison between glaciers based on number of ponds and total ponded area is nonsensical, as these glaciers differ in size and characteristics. Percentage area is a better basis for comparison.

L198-200. Repetition.

L204. These values indicate that you have calculated the average surface slope, which differs considerably from the average surface gradient for debris-covered glaciers (e.g. Quincey et al 2007), which is the actual control on pond incidence (Miles et al 2017 JGlac).

L205. Some confusion of the > and

2001;2012; Miles et al 2016 AGlac; Brun et al 2016; Buri et al 2016; Watson et al 2016;2017; etc). Cliffs and ponds form and evolve together; in some cases an isolated cliff or pond is left, but they often share developmental history due to melt feedbacks and water supply.

L260. There is some confusion with regards to this evolution scheme. 20,000 m2 is not a very large pond. Also the stage 2 of development refers to the development of a feature similar to Spillway Lake, which could then expand up-glacier and develop into an Imja-style proglacial lake.

**Figures and Tables**

F1. These outlines roughly correspond to each glacier's debris covered area only. Also it is possibly worth noting that the Imja Lake and terminal moraine have been included in Imja Glacier's outline, so these aresomehow debris-covered glacier systems?

T1. Possibly worth including Watanabe et al (1986) and even Fritz Muller's work (1962) both of which reported these features.

T2. Again, these characteristics are only for the debris-covered areas of the glaciers.

F2. This excel plot is not very aesthetic, and a poor use of space. Instead of 3 panels, one could combine axes and plot types (e.g. lines or markers) to a single figure.

F3. What do i and ii indicate? Not clear from caption.

F4. Unusual to continue the subpanel numbering from the previous plot.

F5. This is an odd way of visualizing a glacier, as the reference point is actually the middle of the glacier! Better to start from the terminus (or terminal moraine) and work upwards, as this at least is a fixed reference.

F6. Same comment as for F5. These two figures should be combined.

F7 and F8. These should also be combined with F5 and F6 – there is no new analysis

TCD
for the data, just the visual comparison with the velocity data (which could be added to the F5 plots).

F9. I suggest adding a column for the total %ponded area. Also, some consideration needs to be made for your ability to detect cliffs with this methodology, which was not entirely clear.

F10. Many authors would argue that Ngozumpa's Spillway Lake is still too small to be considered stage 3. Imja is certainly stage 3 in 2015 though! For the 2015 data, some consideration needs to be made for the difference in season for comparison purposes. Also, 4 years is a very short time to interpret changes in stage, as there haven't been any profound changes.

F11. This two additional stages are meaningless as they operate continuously on glaciers that exhibit cliffs or ponds, and the authors have provided no evidence that glaciers currently without cliffs and ponds will someday develop them. Rather, early observations on these glaciers did note the presence of cliffs and ponds more or less where we find them today (though likely with lesser frequency).

Suppmat misc

Choice of scene dates is unusual

Figure S1. Units incorrect – these values are considerably larger than the respective glaciers!

Figure S3. These charts should be combined. How were ponds smaller than 100m2 derived? That should not be possible with Sentinel-2 data.

References

Benn, D. I., Thompson, S., Gulley, J., Mertes, J., Luckman, A., & Nicholson, L. (2017). Structure and evolution of the drainage system of a Himalayan debris-covered glacier, and its relationship with patterns of mass loss. Cryosphere.

TCD
Carrivick, J. L., & Tweed, F. S. (2013). Proglacial lakes: character, behaviour and geological importance. Quaternary Science Reviews, 78, 34–52. https://doi.org/10.1016/j.quascirev.2013.07.028

Gardelle, J., Arnaud, Y., & Berthier, E. (2011). Contrasted evolution of glacial lakes along the Hindu Kush Himalaya mountain range between 1990 and 2009. Global and Planetary Change, 75(1–2), 47–55. https://doi.org/10.1016/j.gloplacha.2010.10.003

Haeberli, W., Frauenfelder, R., Hoelzle, M., & Maisch, M. (1999). On rates and acceleration trends of global glacier mass changes. Geografiska Annaler: Series A, Physical Geography, 81(4), 585-591.

Harrison, S., Jeffrey, S. ., Christian, H., John, R., Dan, D. H., Richard, A. ., ... Vít, V. (2018). Climate change and the global pattern of moraine-dammed glacial lake outburst floods. The Cryosphere, 1195–1209. https://doi.org/10.5194/tc-12-1195-2018

King, O., Dehecq, A., Quincey, D., & Carrivick, J. (2018). Contrasting geometric and dynamic evolution of lake and land-terminating glaciers in the central Himalaya. Global and Planetary Change, 167(May), 46–60. https://doi.org/10.1016/j.gloplacha.2018.05.006

Mertes, J. R., Thompson, S. S., Booth, A. D., Gulley, J. D., & Benn, D. I. (2017). A conceptual model of supra-glacial lake formation on debris-covered glaciers based on GPR facies analysis. Earth Surface Processes and Landforms, 42(6), 903-914.

Miles, E. S., Willis, I. C., Arnold, N. S., Steiner, J. F., & Pellicciotti, F. (2017). Spatial, seasonal, and interannual variability of supraglacial ponds in the Lang-tang Valley of Nepal, 1999 to 2013. Journal of Glaciology, (237), 88–105. https://doi.org/10.1017/jog.2016.120

Miles, E. S., Pellicciotti, F., Willis, I. C., Steiner, J. F., Buri, P., & Arnold, N. S. (2016). Refined energy-balance modelling of a supraglacial pond, Langtang Khola, Nepal. AnInteractive comment

**nals of Glaciology, 57(71), 29-40. https://doi.org/10.3189/2016AoG71A421**

Miles, E. S., Steiner, J., Willis, I., Buri, P., Immerzeel, W. W., Chesnokova, A., & Pellicciotti, F. (2017). Pond dynamics and supraglacial-englacial connectivity on debriscovered Lirung Glacier, Nepal. Frontiers in Earth Science, 5, 69.

Quincey, D. J., Richardson, S. D., Luckman, a., Lucas, R. M., Reynolds, J. M., Hambrey, M. J., & Glasser, N. F. (2007). Early recognition of glacial lake hazards in the Himalaya using remote sensing datasets. Global and Planetary Change, 56(1–2), 137–152. https://doi.org/10.1016/j.gloplacha.2006.07.013

Salerno, F., Thakuri, S., D'Agata, C., Smiraglia, C., Manfredi, E. C., Viviano, G., & Tartari, G. (2012). Glacial lake distribution in the Mount Everest region: Uncertainty of measurement and conditions of formation. Global and Planetary Change, 92–93, 30–39. https://doi.org/10.1016/j.gloplacha.2012.04.001

Thompson, S., Benn, D.I., Mertes, J. and Luckman, A., 2016. Stagnation and mass loss on a Himalayan debris-covered glacier: processes, patterns and rates. Journal of Glaciology, 62(233), pp.467-485.

Watanabe, O., Iwata, S. and Fushimi, H., 1986. Topographic characteristics in the ablation area of the Khumbu Glacier, Nepal Himalaya. Annals of Glaciology, 8, pp.177-180.

Watson, C. S., King, O., Miles, E. S., & Quincey, D. J. (2018). Optimising NDWI supraglacial pond classification on Himalayan debris-covered glaciers. Remote Sensing of Environment, 217(September 2017), 414–425. https://doi.org/10.1016/j.rse.2018.08.020

---

## Author Comment (AC1) · 3 Jun 2019

Author response to comments from two reviewers Title: Supraglacial pond evolution in the Everest region, central Himalaya, 2015–2018 Authors: Caroline J. Taylor, J. Rachel Carr MS number: tc-2019-12

Dear Editor,

We are very grateful to C. Scott Watson and an anonymous reviewer for taking the time to review our manuscript and for providing detailed constructive comments. On reflection, and given the sensitive nature of glacier hazards, we accept that the manuscript

requires major revisions. We aim to address the comments of both reviewers to help prepare a substantially revised study to be resubmitted as a new paper, of which we hope will be of value to the wider scientific community.

Caroline Taylor and co-author